# PERSONALLLM:
# TAILORING LLMS TO INDIVIDUAL PREFERENCES

**Thomas P. Zollo**[*]
Columbia University
tpz2105@columbia.edu

**Andrew Wei Tung Siah**[*]
Columbia University
andrew.siah@columbia.edu

**Naimeng Ye**
Columbia University
ny2336@columbia.edu

**Ang Li**
Columbia University
al4263@columbia.edu

**Hongseok Namkoong**
Columbia University
hongseok.namkoong@columbia.edu

## ABSTRACT

As LLMs become capable of complex tasks, there is growing potential for personalized interactions tailored to the subtle and idiosyncratic preferences of the user. We present a public benchmark, PersonalLLM, focusing on adapting LLMs to provide maximal benefits for a particular user. Departing from existing alignment benchmarks that implicitly assume uniform preferences, we curate open-ended prompts paired with many high-quality answers over which users would be expected to display heterogeneous latent preferences. Instead of persona-prompting LLMs based on high-level attributes (e.g., user race or response length), which yields homogeneous preferences relative to humans, we develop a method that can simulate a large user base with diverse preferences from a set of pre-trained reward models. Our dataset and generated personalities offer an innovative testbed for developing personalization algorithms that grapple with continual data sparsity—few relevant feedback from the particular user—by leveraging historical data from other (similar) users. We explore basic in-context learning and meta-learning baselines to illustrate the utility of PersonalLLM and highlight the need for future methodological development. Our dataset is available at `https://huggingface.co/datasets/namkoong-lab/PersonalLLM`.

## 1 INTRODUCTION

The *alignment* of LLMs with human preferences has recently received much attention, with a focus on adapting model outputs to reflect universal population-level values. A typical goal is to take a pre-trained model that cannot reliably follow complex user instructions (Wei et al., 2022) and can easily be made to produce dangerous and offensive responses (Perez et al., 2022), and adapt it to the instructions of its user base (Ouyang et al., 2022) or train a generally helpful and harmless assistant (Bai et al., 2022). By assuming a *uniform preference* across the population, recent successes (Ziegler et al., 2020; Ouyang et al., 2022; Christiano et al., 2017) demonstrate the feasibility of learning and optimizing a monolithic preference ("reward model"). Alignment techniques have provided the basis for popular commercial applications like ChatGPT, as well as instruction-tuned open-source models (Touvron et al., 2023).

The rapid advancement in LLM capabilities opens the door to an even more refined notion of human preference alignment: personalization. A personalized model should adapt to the preferences and needs of a particular user, and provide maximal benefits as it accumulates interactions (see Figure 1). Given the expected data sparsity in this setting, such personalized language systems will likely also rely on historical data from other (similar) users to learn how to learn from a small set of new user feedback (see Figure 2). By discovering patterns across users, these systems can efficiently optimize their responses, ultimately leading to more effective and beneficial conversational AI. For example, personalized learning experiences could be crafted by adapting educational chat assistants to an individual student's specific learning pace and style based on previous successful interactions with similar students. Customer support chatbots could offer more accurate and empathetic responses by drawing on a wealth of previous interactions, leading to quicker resolution times and higher customer satisfaction. In

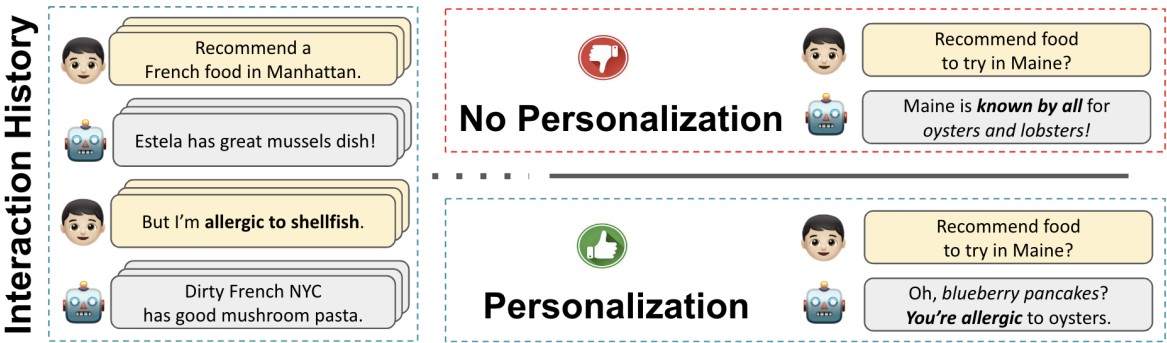

Figure 1: Standard LLMs require tedious re-prompting to learn a user's preferences in each session. Personal-LLM aims to learn a unique user's diverse preferences to maximize long-term satisfaction.

healthcare, personalized chatbots could provide tailored advice based on patients with similar medical histories and communication preferences.

Compared to conventional applications where prompts have a uniform notion of "ground truth" (e.g., question answering), LLM personalization is distinguished by the need to study open-ended prompts where users exhibit heterogeneous preferences across many possible high-quality answers (Figure 1). While personal preferences may vary according to simple features like user age Chan et al. (2024); Castricato et al. (2024) and answer length and technicality Li et al. (2024), they also involve more abstract dimensions of culture, politics, and language Kirk et al. (2024), as well as aspects of personality that are difficult to explain (Hwang et al., 2023). A personalized LLM should be able to adapt to subtle, idiosyncratic, and sometimes sensitive differences between user tastes as it gathers more interactions.

Inspired by the vision of a future with personalized AI, we introduce PersonalLLM, a public, open-source benchmark designed to adapt LLMs to provide maximal benefits for individual users. In order to explore complex differences in user tastes, our benchmark features a set of prompts with many high-quality LLM responses (from state-of-the-art LLMs like GPT-4o, Claude 3 Opus, and Gemini 1.5 Pro), such that humans *are expected* to express diverse preferences over the responses. Such an approach to dataset-building stands in contrast to existing alignment datasets, where responses exhibit observable quality differences (see Figure 3).

For each prompt and set of responses, our dataset also includes scores from a set of 10 reward models with heterogeneous preferences over those responses. We leverage these reward models to sample many synthetic "users" (or personal preference models) via weighted ensembles of their preferences, and in doing so we are able to *simulate an entire user base*, which we argue to be a critical ingredient in a truly useful personalization benchmark. Through extensive analysis of the preferences of these users over our dataset, we show these simulated personal preference models to be diverse and non-trivial (e.g., with respect to length, formatting, or tone). We illustrate the difficulty of creating such an environment by comparing to the increasingly popular persona prompting baseline (Castricato et al., 2024; Chan et al., 2024; Jang et al., 2023), which in our analysis produces preferences only half as diverse as a set of PersonalLLM users across multiple metrics. Taken together, the prompts, responses, and personalities present in PersonalLLM offer an innovative test for benchmarking personalization algorithms as they tailor interactions based on previous interactions with an individual user.

While fine-tuning and reinforcement learning approaches (Schulman et al., 2017; Rafailov et al., 2023) are effective for aligning to population-level preferences, personalization requires a new algorithmic toolkit, as it is not practical to gather enough data or store a separate copy of the model or even low-rank adapter weights (Hu et al., 2021) for every user. PersonalLLM offers the versatility necessary to spur development across a range of new approaches to personalization: in-context learning (ICL) (Brown et al., 2020), retrieval augmented generation (RAG) (Lewis et al., 2021a), ranking agents, efficient fine-tuning, and other adaptation techniques. In our experiments, we highlight a particularly salient challenge compared to typical applications of personalization in recommendation systems: the space of "actions/responses" is prohibitively large to be able to explore based on interactions with a single user. Since this necessitates *learning across users*, we model this as a meta-learning problem where the goal is to leverage a wealth of prior interactions from historical users to tailor responses for a new user who does not have a significant interaction history (see Figure 2).

Motivated by key methodological gaps in personalizing LLMs, here we summarize our contributions: 1) We release a new open-source dataset with over 10K open-ended prompts paired with 8 high-quality responses from

Figure 2: PersonalLLM enables the development of methods for learning *across* users, where a dataset of historical users and their interactions is leveraged to personalize interactions for a new user with a limited history.

top LLMs. 2) We propose a novel method for sampling "users" (i.e., personal preference models) that, unlike existing methods, creates diverse preferences and allows for the simulation of large historical user bases. 3) We illustrate new possibilities for algorithmic development in learning *across* users.

Our goal in creating the open-source PersonalLLM testbed is to facilitate work on methods to personalize the output of an LLM to the individual tastes of many diverse users. We do not claim that our simulated personal preference models provide a high-fidelity depiction of human behavior. Instead, they offer a challenging simulation environment that provides the empirical foundation for methodological innovation in capturing the complex array of human preferences that arise in practice. As an analogy, while ImageNet (Russakovsky et al., 2015) is noisy and synthetic—e.g., differentiating between 120 dog breeds is not a realistic vision task—it provides a challenging enough setting that methodological progress on ImageNet implies progress on real applications. Similarly, we believe PersonalLLM is a reasonable initial step toward the personalization of language-based agents, building on the common reinforcement learning paradigm of benchmarking personalization algorithms with simulated rewards (Zhao et al., 2023; Ie et al., 2019).

## 2 PERSONALLLM

Our PersonalLLM testbed is composed of two high-level components: 1) a dataset of prompts, each paired with a set of high-quality responses among which humans would be expected to display diverse preferences and 2) a method for sampling diverse personal preference models, such that we can test methods for personalization using these "personas" as our simulated users. Next, we will describe each of them in detail. Our data [1] and code [2] are publicly available, and full documentation for our dataset is available in Appendix A.

### 2.1 DATASET

Since our goal is to study diverse preferences, we first focus on collecting *open-ended* prompts, similar to a chat setting. We compile 37,919 prompts from Anthropic Helpful-online, Anthropic Helpful-base (Bai et al., 2022), Nvidia Helpsteer (Wang et al., 2023), and RewardBench (Lambert et al., 2024). From this set, prompts are filtered to those with a length of 2400 characters or fewer as most reward models are limited to 4096 context length. We then randomly draw 10,402 prompts to form our final set.

Our next aim is to collect many high-quality responses for each prompt. Important desiderata for the generated responses are that i) they do not exhibit much variation in terms of undesirable contents (like misinformation or toxicity) or obvious dimensions of helpfulness or length, as is typical in RLHF datasets, ii) they exhibit diversity across meaningful dimensions of personal preferences like political viewpoint and culture, as well as difficult to describe latent features. To achieve this, we generate eight responses for each of these 10,402 prompts using a selection of the top models from ChatArena and other important benchmarks: **GPT-4o, Claude 3 Opus, Gemini-Pro-1.5, Command-R-Plus, GPT-4-Turbo, Claude 3 Sonnet, Llama3-70B-Instruct, and Mixtral**

---

[1] https://huggingface.co/datasets/namkoong-lab/PersonalLLM
[2] https://github.com/namkoong-lab/PersonalLLM

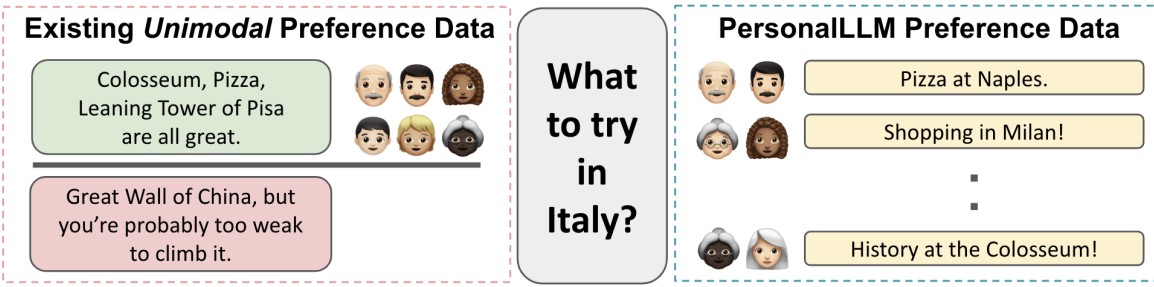

Figure 3: **Left:** Existing alignment datasets contain prompts paired with multiple responses, where the majority of people are expected to prefer one specific response (e.g., a harmless response). **Right:** Our dataset consists of prompts paired with many high-quality responses, creating a testbed to build PersonalLLMs.

**8x22B**. We split the resulting dataset into 9,402 training examples and 1,000 test examples. See Appendix Figure 7 for an illustration of our dataset construction pipeline.

## 2.2 SIMULATING PERSONAL PREFERENCE MODELS

We design our approach to creating simulated PersonalLLM users with several goals in mind. First, we aim for PersonalLLM to allow for the simulation of a large number of users, enabling the study of the full personalization paradigm for applications such as search engines and recommender systems (Davidson et al., 2010; Das et al., 2007; Xu et al., 2022; Färber and Jatowt, 2020) wherein a historical database of user data is leveraged to personalize new interactions. Next, when applied to our dataset, our preference models should allow for the study of alignment based on diverse and complex latent preferences, as opposed to simple attributes such as answer length or sensitive and reductive user characteristics, for example, race or gender. Finally, our evaluation should not rely on GPT4, which can be expensive and unsuitable for research purposes given model opacity and drift. While human annotations are the gold standard ideal for preference evaluation (Kirk et al., 2024), it is often impractical to obtain this feedback consistently throughout the development cycle. As a result, we claim that synthetic personal preference models are needed to facilitate progress in LLM personalization.

To overcome this difficult challenge of simulating diverse preferences, we propose a solution based on a set of strong open-source RLHF reward models. While it may be the case that different reward models have fairly uniform preferences over the high-quality/low-quality response pairs on which they are typically trained, we hypothesize that their preferences over many high-quality responses will instead be diverse. Since the number of existing top-quality reward models is much smaller than the number of users we would like to simulate, we propose to generate users by sampling weightings over the set of reward models, such that the reward score assigned to a (prompt, response) pair by a user is a weighted sum of the reward scores assigned by the pre-trained reward models. In Section 3, we validate our hypothesis regarding the diverse and non-trivial preferences created by such sampling.

More formally, for an input prompt $x \in \mathcal{X}$, an LLM produces output response $y \in \mathcal{Y}$, where $\mathcal{X}$ and $\mathcal{Y}$ are the set of all-natural language. Then, a preference model $R : \mathcal{X} \times \mathcal{Y} \to \mathbb{R}$ assigns a reward score to the response given to the prompt, with higher scores indicating better responses. Next, consider a set of $B$ base reward models, denoted as $RM_b$, $b = 1, \ldots, B$, and a set of $k$ $B$-dimensional weightings, which represent a set of personal preference models. The preference model corresponding to user $i$ can then be defined by a weighted average of these $B$ base models $RM_1, RM_2, \ldots, RM_B$, with weights $w_1, w_2, \ldots, w_B$: $R^i(x, y) = \sum_{b=1}^{B} w_b^i \cdot RM_b(x, y)$. For our base reward models $\{RM_b\}_{b=1}^{B}$, we select 10 reward models with strong performance on RewardBench, an open-source benchmark for evaluating reward models. These reward models are built on top of popular base models such as Llama3, Mistral, and Gemma (see Appendix A). We evaluate each (prompt, response) pair in the train and test set with each model so that for any personality created in this manner, each (prompt, response) pair in the dataset can be scored via a simple weighting. Together, our dataset and personal preference models create an innovative and challenging environment for developing personalization methodology. This extends the existing paradigm of simulated rewards, commonly used in domains like recommender systems (Zhao et al., 2023; Ie et al., 2019), to the task of LLM personalization.

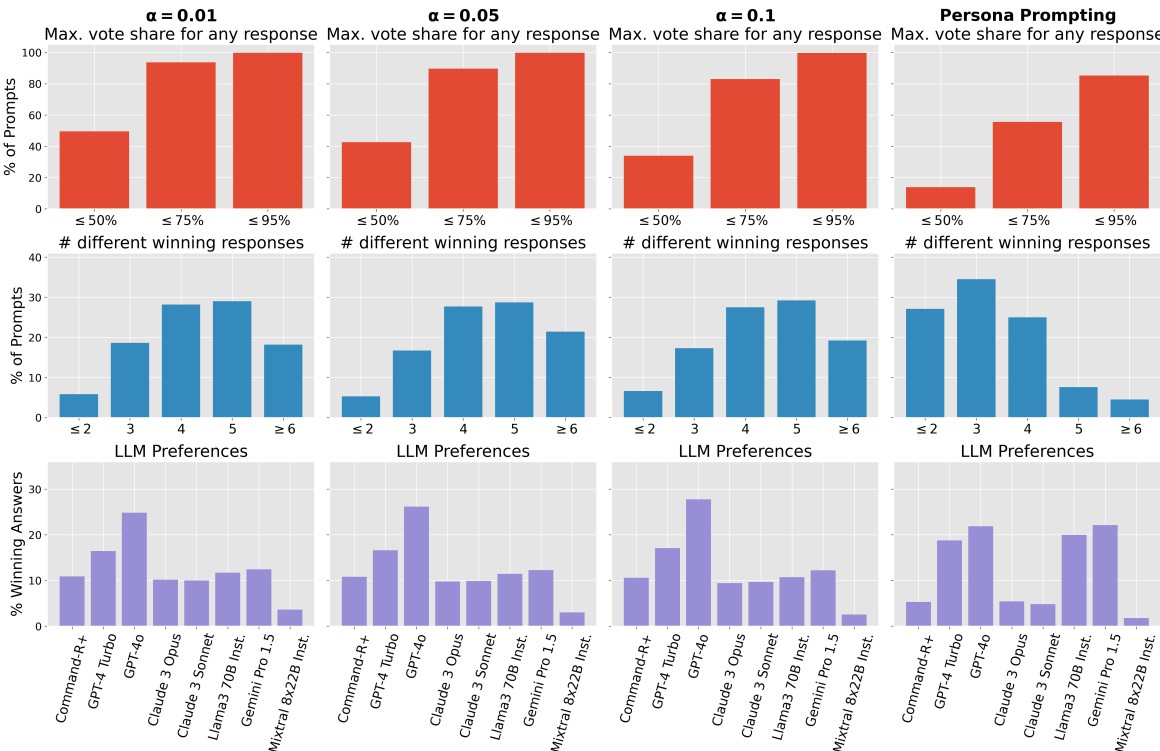

Figure 4: Probing the heterogeneous preferences of our simulated users across the PersonalLLM dataset compared to a persona prompting baseline. **Top**: For a population of simulated users, the percentage of each population's vote share given to the most common winning response for each prompt. **Middle**: A histogram showing the number of responses that receive at least one vote from a simulated population for each prompt. **Bottom**: Average win rates across the population for the 8 LLMs in our dataset.

### 2.2.1 SAMPLING USER WEIGHTINGS

There are many valid ways to sample the $B$-dimensional weighting vectors. As a simple starting point, we propose to sample from a Dirichlet distribution with a uniform concentration parameter across all classes ($w \sim \text{Dirichlet}(\alpha)$). As $\alpha$ becomes very small, the preference models converge towards the 10 base reward models; as it becomes large, preferences become unimodal. Such a parameter allows us to simulate user bases with different underlying preference structures, as we detail in the next section.

## 3 ANALYZING PERSONALLLM

Next, in order to validate our testbed, we explore the preferences exhibited by our simulated users over the PersonalLLM dataset.

**Preference Diversity and Comparison to Persona Prompting** First, we examine whether populations of personal preference models sampled via the method outlined in Section 2.2 do in fact display heterogeneous preferences over the prompt/response pairs in our dataset. In Figure 4 (left 3 columns), we provide experimental results for user bases of 1,000 PersonalLLM personal preference models sampled with parameters $\alpha = [0.01, 0.05, 0.1]$ and applied to the PersonalLLM test set to choose winning responses among the 8 included. The top row displays the percentage of prompts in the dataset for which the most popular winning response according to the population receives no more than 50%, 75%, and 95% of the population vote; higher values indicate more diversity in preferred responses. The middle row shows the percentage of prompts that have a given number of responses with at least one winning vote across the population; heterogeneous population preferences induce higher concentration on the right side of each plot. On the bottom, we plot the overall win rates for each LLM across all users and prompts.

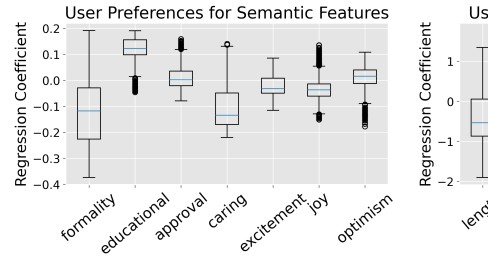 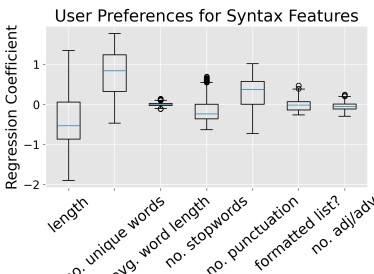 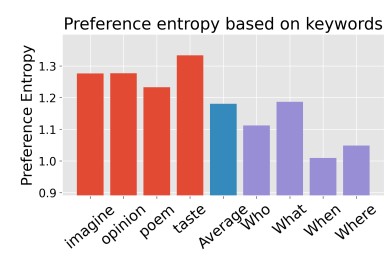

Figure 5: Simulated user preferences with respect to prompt and response contents. **Left, middle:** For each user, a regression model predicts winning responses based on semantic (left) or syntactic (middle) features. **Right:** We examine the entropy in population preferences based on keywords in prompts, comparing words we expect to inspire heterogeneity (e.g., imagine, poem) to prompts beginning with "who", "when", and "where".

In the right column, we also offer results for a persona prompting baseline. Persona prompting (Castricato et al., 2024; Chan et al., 2024; Jang et al., 2023) is an emerging method for evaluating methods for LLM personalization, wherein an LLM is prompted to decide which response would be preferred by a person of a particular race, gender, age, profession, or other demographic category. While one might argue that such evaluation is *prima facie* discriminatory and reductive, and therefore not a desirable standard for algorithmic advancement, especially in sensitive areas, we are also interested in whether persona prompting meets the technical challenge of producing a simulation environment with a high degree of heterogeneity. For our baseline, we prompt the sfairXC/FsfairX-LLaMA3-RM-v0.1 reward model (Dong et al., 2023) to score responses with respect to 1,000 personas randomly sampled from PersonaHub Chan et al. (2024), a recent effort at building a database of personas that are representative of a pluralistic population.

Results are shown in Figure 4. First, we observe the left-most bar in each plot in the top row. We can see that for PersonalLLM personas the top response receives a majority user vote for only about half of the prompts, while that figure is closer to 90% for the persona prompting baseline. By summing the right-most two bars in each plot across the middle row, one can observe that for roughly 60% of prompts, at least 5 different answers are chosen as the best by at least 1 user under our set of personas. For LLM persona prompting, it is roughly 30%. Finally, our ensembled preference models have a fairly diffuse set of preferences over the response-generating LLMs, while persona prompting strongly prefers a subset of 4 models. With respect to changes across the left 3 columns, from the top row of plots we can observe that as $\alpha$ increases, preferences become more uniform, with more prompts featuring one highly preferred response. However, if $\alpha$ is set too low, user preferences cluster very tightly around the base reward models (see Appendix Figure 9).

**Effects of Semantics and Syntax** We further analyze the effects of semantics and syntax on the preferences of a simulated user base (with $\alpha = 0.05$ and 1,000 users). We use regression analysis to understand how different features may drive the preferences of different users, including semantic response features such as the formality or educational value or the expressions of certain emotions (approval, caring, excitement, joy, optimism), as well as syntactic features like length and the use of different parts of speech and formatting. For each user, we gather their most and least preferred responses for each of the test prompts and create a binary prediction problem to predict whether a given response is a winning or losing response. Responses are embedded using hand-crafted features (based on either syntax or semantics, which are studied separately), and a unique logistic regression model is trained *for each user*. Semantic features were captured using pre-trained classifiers, while syntactic features were engineered using nltk (Bird and Loper, 2004). See Appendix B.1 for complete details.

In Figure 5 (left and middle), for each feature we show a box plot with the resultant regression coefficient for that feature across users. A positive coefficient suggests a feature associated with winning responses, while a negative coefficient suggests a feature's role in losing responses. A tight box indicates homogeneous preferences, while a greater spread represents heterogeneity. Here, we can see a reasonable mix of heterogeneity and homogeneity across user preferences for different features. Semantically, users tend to prefer responses with educational value and dislike highly formal responses, although the size of these preferences varies. Encouragingly, syntactic preferences do not seem to be driven by uniform preferences for simple features like length or the presence of formatting such as bullets or lists. In Figure 5 (right), we compare the Shannon entropy (Shannon, 1948) in the population preferences over the responses to a given prompt based on keywords. By comparing words we would expect to inspire heterogeneity (e.g., "imagine", "opinion", "poem") to prompts beginning with "who", "when", and "where", which evoke more objective answers. We can see that the presence of these subjective

Table 1: Representativeness scores in relation to real human opinions from important demographic groups for different LLMs, as well as our PersonalLLM population.

| Demographic | AI21 Labs | | | OpenAI | PersonalLLM |
| | j1-jumbo | j1-grande-v2 | ada | text-davinci-003 | **Ours** |
| --- | --- | --- | --- | --- | --- |
| Black | 0.820 | 0.812 | 0.823 | 0.702 | **0.833** |
| White | 0.807 | 0.794 | 0.817 | 0.699 | **0.832** |
| Less than $30,000 | 0.828 | 0.813 | 0.833 | 0.693 | **0.838** |
| $100,000 or more | 0.797 | 0.790 | 0.807 | 0.708 | **0.831** |

cues in prompts leads to a more diverse set of preferences than those seeking simple entity or date responses. Such diversity among the prompts creates a setting where an algorithm *must not only learn how to personalize but also when to personalize*. We extend the same analysis to persona prompting in Appendix Figure 8.

**Comparison to Human Preferences** Finally, to understand how our synthetic personal preference models relate to human preferences over text responses, we surveyed a population of our simulated users on a set of questions with responses where a large and diverse set of humans have given their preferences in the past, the OpinionQA dataset (Santurkar et al., 2023). OpinionQA is an appropriate validation set for our personas given that its broad coverage of topics (e.g., science, economics, politics, romance, and many other topics) aligns with the open-domain nature of our prompt set. See Appendix Table 3 for example questions and answers. Following previous work, we calculate the representativeness score of the opinion distribution given by our simulated preference models. This score is based on the Wasserstein distance of the synthetic population preferences from that of a real human population for each question.[3] To have a high representativeness score, our simulated user population would have to display heterogeneous preferences over question/response sets where humans do so and produce homogeneous (and matching) preferences in cases where humans do the same. Our population of simulated users produces a representativeness score of 0.839 with respect to the overall population of the US, higher than any LLM in the original study and near as representative of the overall population as some real, large demographic groups. Further, in Table 1 we can see that our simulated users produce opinions that better represent a wide range of important (and sometimes protected) groups according to demographic attributes such as race and income. In fact, this is the case for 59 of 60 demographic groups studied (see Appendix Table 4).

**Summary of Analysis** These results show that our simulated user reward models: 1) produce heterogeneous preferences over our dataset, considerably more so than persona prompting an LLM, 2) display reasonable and diverse preferences with respect to the syntactic and semantic content of prompts, and 3) simulate a user base that better represents diverse human opinions than many popular LLMs, without resorting to explicit stereotyping.

## 4  PERSONALIZATION EXPERIMENTS

A central challenge in personalization is the perpetual lack of data, as most users will provide sparse feedback, far less than necessary to effectively adapt an LLM. Two problems emerge from such an environment: 1) how to best leverage small amounts of user-specific data for personalized adaptation and 2) how to lookup similar users based on language feedback. In order to illustrate how researchers might approach these problems, we perform experiments in two modal settings for LLM personalization research. First, we explore a scenario where we have access to a short but relevant interaction history (i.e., previous prompts and preference feedback) for the user, and we aim to efficiently leverage that interaction history through ICL. Then, we explore a more complex setting that fully leverages the advantages of PersonalLLM, where the current user possibly has no relevant interaction history, and we must instead retrieve relevant interactions from similar users in a database. Overall, our results validate the solid empirical foundations of PersonalLLM while highlighting salient algorithmic questions and the fact that there is much room for improvement in terms of personalization performance.

All experiments simulate a chatbot using in-context learning to personalize responses for a test set of new users. Our test set simulates 1,000 personal preference models (or "users") drawn with $\alpha = 0.05$ (as in Section 3), and each user is associated with one test prompt from the PersonalLLM test split. For a new user with an associated

---

[3]The score is $1 - \mathcal{W}$ for distance $\mathcal{W}$; a higher score indicates more representative preferences.

test prompt, the goal is to use ICL to produce a response to maximize the reward (and win rate vs. GPT4o) given by the user's personal preference model (i.e., weighted ensemble of reward models). Our underlying chatbot is Llama3-8B-Instruct, and all text embeddings are extracted using the top-ranking model on the MTEB (Muennighoff et al., 2023) leaderboard below 500M parameters.[4] Further details are given in Appendix C.1.

## 4.1 PERSONALIZED IN-CONTEXT LEARNING

While ICL for broad alignment has been studied to some extent (Lin et al., 2023), the problem may be different when the underlying preference model is idiosyncratic and may cut against pretraining and RLHF dataset biases. In our initial set of experiments, we focus on a setting wherein we have a small set of useful data for the sake of personalizing the response to a given query, i.e., feedback gathered from the same user on similar prompts. By doing so, we can study key questions related to personalized inference with ICL, for example, which response(s) should be included and the importance of correct labels. Though these questions have been studied with respect to more general NLP tasks (Min et al., 2022; Yoo et al., 2022; Pan et al., 2023), it is unlikely that these findings can be extrapolated to the unique personalization task, and thus more work is needed in this area. A solid foundation of ICL techniques for personalization can then form the basis for more complex systems involving, e.g., looking up similar users.

**Experiment Details** For each of our 1,000 test users, each with their own test prompt, we build a short but relevant interaction history by retrieving 5 other prompts based on embedding similarity. We build a winning/losing response pair for each prompt based on each user's most and least preferred answers from the 8 models in our dataset. In order to establish baseline results on key questions in personalization, we include several baselines for how these interaction samples are leveraged in-context during inference: 1) **Winning and Losing:** Both the winning and losing responses are included. 2) **Winning only:** Only the winning response is included. 3) **Losing only:** Only the losing response is included. 4) **Losing only (Mislabeled):** Only the losing response is included, mislabeled as a winning response. Inference is performed using 1, 3, and 5 such examples (see Appendix C.1 for exact templates), and evaluated by scoring with each user's (weighted-ensembled) preference model. We also compare to a zero-shot baseline, with no personalization.

**Results** Results are shown in Figure 6, left column. We can see that the best performance comes from ICL with only winning examples. This underlines the outstanding challenge of training LLMs to not only mimic winning responses in-context, but also leverage the contrast between winning and losing responses, especially when the differences may not described in the model's training data. Any amount of examples, even incorrectly labeled, are helpful relative to zero-shot; this may be unsurprising, as all 8 models in our dataset are stronger than our 8B parameter chat model. An interesting result lies in the comparison between Losing Only and Losing Only (Mislabeled). While the mislabeled examples may help performance versus a zero-shot baseline (once again because they are from a stronger underlying LLM), Llama-8B-Instruct gains more from having these relatively strong losing responses labeled as losing. Overall, our findings reflect that a model trained for broad alignment does have some of the necessary capabilities to do idiosyncratic personalization using only in-context examples, but that much work is left in order to fully leverage this language feedback.

## 4.2 LEARNING ACROSS USERS

Having established some empirical foundations for in-context personalization with PersonalLLM, we next highlight a particularly significant challenge prevalent in practice that has been under-explored in the LLM community: the cold-start problem. When a new user with limited prior interaction data arrives, or a user inquires about a new topic, prior user interactions alone cannot inform a satisfactory response. We model this challenge as a meta-learning problem, where the goal is to utilize a rich reservoir of prior interactions with a diverse set of users. We are motivated by real-world scenarios where we have access to a proprietary database containing extensive interaction histories from previous users. When a new user arrives, our goal is to utilize this rich, heterogeneous dataset to provide the best possible response to the new user's query despite having only limited initial interactions with them that may not be relevant to the current query. This setting resembles typical recommendation systems, but "actions" are now defined over the space of natural language outputs instead of a fixed set of items. See Figure 2 for further illustration.

---

[4]https://huggingface.co/dunzhang/stella_en_400M_v5

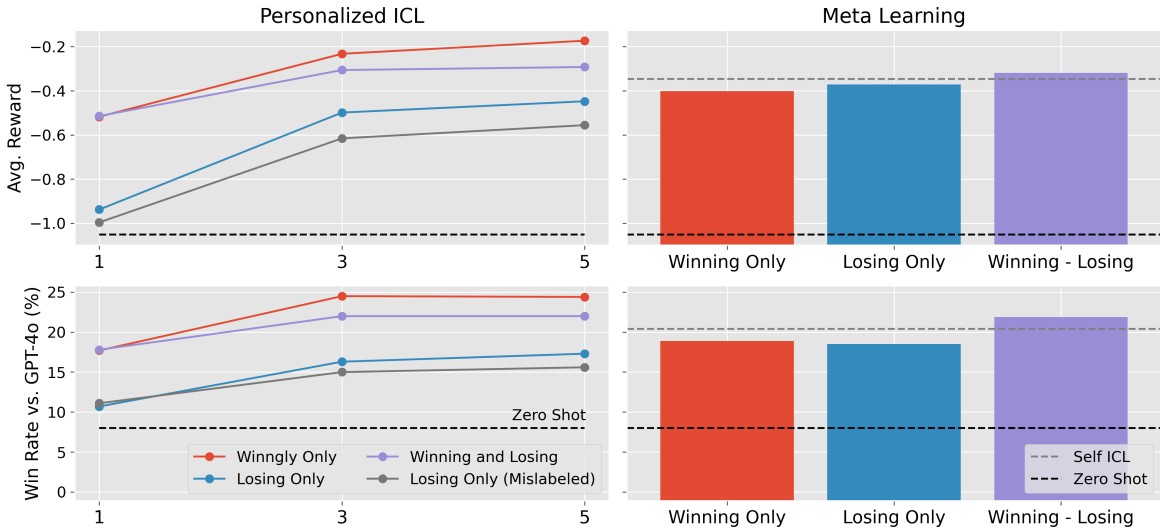

Figure 6: Results across different personalization algorithms. **(Left)** Test users are accompanied by a relevant interaction history with pairwise preference feedback, and we explore the LLM's ability to exploit this information in context. **(Right)** Test users have interaction histories that are not relevant to their test prompt, and we probe methods for embedding users based on language feedback to retrieve useful examples from similar users.

Our experiment explores the open question of how best to embed (i.e., represent with some vector) users based on small amounts of natural language feedback. With effective algorithms to lookup similar users, more relevant interactions may be leveraged to improve a response to a new user query. While a rich literature exists on information retrieval (i.e., RAG) for typical NLP benchmark tasks like question answering and fact-checking (Lewis et al., 2021b; Gao et al., 2024), the personalization task necessitates new algorithms.

**Experiment Details**  For each of our 1,000 test users, we build a short but, in contrast to our first experiment, *possibly irrelevant* interaction history by retrieving 5 random prompts. Winning/losing response pairs (i.e., preference feedback) are selected as before. In order to supplement these interaction histories, we sample a historical database of 10,000 users (also with $\alpha = 0.05$), each with a set of 50 prompts, winning response, losing response triplets from the train set, where the prompts are selected randomly and the winning and losing responses are selected as the historical user's highest and lowest scoring among the 8. We compare 3 methods for embedding users for lookup: 1) **Winning minus Losing:** Average direction in embedding space between winning and losing responses for each prompt. 2) **Winning only:** Average direction in embedding space for winning responses. 3) **Losing only:** Average direction in embedding space for losing responses. For each test user, we build a set of candidate prompt/feedback data by retrieving the 20 most similar historical users based on the cosine similarity of their embeddings, and then of the pool created by those users' interaction histories, retrieving 3 examples for in-context learning based on prompt embedding similarity to the test user's new query. We compare to a **Self-ICL** baseline, where the test user's possibly irrelevant prompt/feedback history is used for ICL. All methods use only winning responses in-context; evaluation is done as before.

**Results**  Our results are shown in Figure 6. We find that using the strongest user embedding method, which most fully exploits the available pairwise preference feedback, meta-learning can beat the self-ICL baseline. This positive result for meta-learning highlights the opportunity created by leveraging historical user data, and the feasibility of embedding users based on a small amount of language feedback. However, the gain from our relatively naive method is small, illustrating the need for methodological innovation in building such systems.

## 5  RELATED WORK

**Preference Datasets**  Recent developments in large language models (LLMs) emphasize the importance of *aligning* LLMs based on *preference feedback* rather than merely pre-training on large corpora of language in a self-supervised manner. Consequently, there has been a surge in the creation of open-source datasets (Bai et al., 2022; Nakano et al., 2022; Köpf et al., 2023; Dubois et al., 2024; Lambert et al., 2024) designed to support research on alignment methodologies. A significant limitation in the existing datasets is that they mainly

enable fine-tuning to a single high-level notion of alignment that is uniform across the population, such as instruction-following in RLHF (Ouyang et al., 2022) and helpfulness and harmlessness (Bai et al., 2022).

**Personalization** Personalization has been extensively researched across different fields, with previous datasets primarily focusing on applications such as search engines and recommender systems (Davidson et al., 2010; Das et al., 2007; Xu et al., 2022; Färber and Jatowt, 2020). Where language model personalization has been studied, it has typically been focused on learning to mimic a user's style, for example in writing headlines, crafting social media posts, or emulating historical text with dialogue models (Salemi et al., 2024; Vincent et al., 2023; Welch et al., 2022; Ao et al., 2021; Wu et al., 2021; Mazaré et al., 2018). Recently, given the success of population-level alignment, researchers have begun to develop testbeds and methodology wherein the goal is to achieve a more granular level of personalized alignment for LLMs (Castricato et al., 2024; Jang et al., 2023; Kirk et al., 2024; Li et al., 2024). Much of this work has focused on alignment for real or synthetic personas based on high-level attributes like race or occupation (Castricato et al., 2024; Chan et al., 2024), or high-level notions of alignment with respect to response qualities like length, technicality, and style. For example, Jang et al. (2023) decomposes personal preferences along a handful of easily observable dimensions and performs personalized generation by merging models trained with different preference data based on these dimensions. Evaluation is often done by prompting GPT4 to select the preferred response based on preferences stated in its prompt (Jang et al., 2023; Castricato et al., 2024). In an effort to highlight the need for broad participation and representation in LLM alignment, the PRISM dataset collects user-profiles and personalized preference feedback from over 1,000 diverse human participants (Kirk et al., 2024).

# 6 DISCUSSION

**Meta-Learning for Personalization** We hope to encourage more work in the meta-learning setting, as exemplified by our experiments. This setting mirrors many real-world use cases where an organization has a large proprietary dataset from historical users but a very limited interaction history with this particular user. Prior work on cold-start problems has focused on the task of recommending discrete content items from a media (or other) library. Extending and developing these techniques for LLMs is an exciting direction for future research.

**Risks and Limitations** We must consider the risks and limitations associated both with the release of our original benchmark dataset, as well as the larger goal of LLM personalization. With respect to PersonalLLM, we note all prompts and responses have not been manually inspected for quality or safety by a human, although prompts are sourced from existing, reputable datasets, and responses are generated from state-of-the-art language models that have (presumably in the case of black box models) undergone safety alignment. Our benchmark is also limited with respect to the realism of the personas created by weighting reward models. On a broader note, the goal of LLM personalization brings particular risks, including filter bubbles, stereotypes, feedback loops, personification risks, and user manipulation. Given these and many other predictable (and unpredictable) potential risks, it is important that any efforts at LLM personalization are accompanied by research in robust transparency mechanisms and safeguards for personalization algorithms. Developing an empirical foundation for such efforts is another promising avenue for future work.

**Future Directions** Given that LLMs have only recently reached a level of capabilities meriting their widespread adoption for industrial and personal use, the study of LLM personalization is necessarily in its earliest stages of development. It follows that there are many important and exciting avenues for future research, with respect to datasets, methodology, fairness, safety, and other aspects of responsible and reliable machine learning deployment. Since PersonalLLM is the first dataset to enable the study of complex personalized preferences expressed over many high-quality responses (to our knowledge) by a large, diverse user base, the benchmark can be extended in many ways. For example, one might imagine a distribution shift scenario, where over time, personal preferences shift, and the personalization algorithm must balance stability and plasticity. Also, given concerns regarding the realism of our simulated users, we hope that our testbed drives the development of even more realistic personalization datasets and evaluation methods that more closely mirror the online and non-i.i.d. settings and better capture the true nuance and diversity of human personal preferences. Finally, continued work in personalization algorithms must be accompanied by work in personalization safety, fairness, and reliability. Future research may consider different aspects of the deployment pipeline (e.g., model architecture, data collection) and interaction model (e.g., UI/UX) with these concerns in mind.

ACKNOWLEDGMENTS

We thank ONR Grant N00014-23-1-2436 for its generous support. This work is supported by the funds provided by the National Science Foundation and by DoD OUSD (R&E) under Cooperative Agreement PHY-2229929 (The NSF AI Institute for Artificial and Natural Intelligence).

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

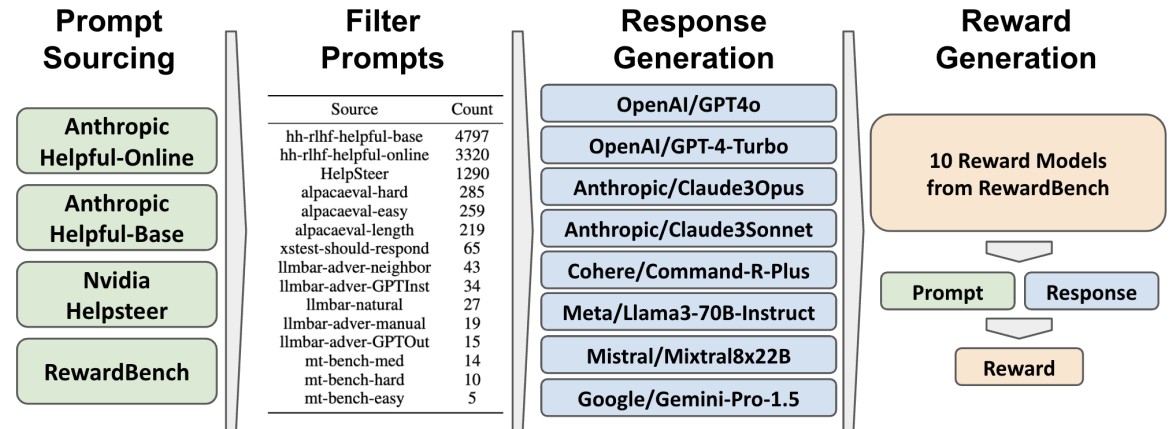

Figure 7: Illustration of our dataset creation pipeline. (Left) Prompts are sourced from various open-source RLHF datasets. (Left, middle) Prompts are filtered by length. (Right, middle) Responses to each prompt are generated by 8 high-quality LLMs. (Right) For each response for each prompt, rewards are scored by 10 different top-performing reward models.

## A   DATASET INFORMATION

This section serves as documentation for our dataset, with its organization based on the notion of datasheets (Gebru et al., 2021).

Our code and dataset will be made publicly available upon the release of this paper.

### A.1   COMPOSITION

#### A.1.1   PROMPTS

In order to create a set of prompts and responses over which humans (and reward models) will display diverse preferences, our first focus was the collection of *open-ended* prompts. As a source of these open-ended prompts, we collected data from **Anthropic Helpful-online**, **Anthropic Helpful-base**, **Nvidia Helpsteer**, and **RewardBench**. From this set, prompts were filtered to those with a length of 2400 characters or fewer as most reward models are limited to 4096 context length. We randomly drew 10,402 prompts from the filtered subset. The resulting distribution of prompts from different sources is shown in Table 2.

Table 2: Sources of the 10,402 prompts composing our train and test sets.

| Source | Count |
| --- | --- |
| hh-rlhf-helpful-base | 4797 |
| hh-rlhf-helpful-online | 3320 |
| HelpSteer | 1290 |
| alpacaeval-hard | 285 |
| alpacaeval-easy | 259 |
| alpacaeval-length | 219 |
| xstest-should-respond | 65 |
| llmbar-adver-neighbor | 43 |
| llmbar-adver-GPTInst | 34 |
| llmbar-natural | 27 |
| llmbar-adver-manual | 19 |
| llmbar-adver-GPTOut | 15 |
| mt-bench-med | 14 |
| mt-bench-hard | 10 |
| mt-bench-easy | 5 |

### A.1.2 RESPONSES

Next, we aimed to collect many high-quality responses for each prompt. We generated eight responses for each of the 10,402 prompts using a selection of the top models from ChatArena and other important benchmarks: **GPT-4o**, **Claude 3 Opus**, **Gemini-Pro-1.5**, **Command-R-Plus**, **GPT-4-Turbo**, **Claude 3 Sonnet**, **Llama3-70B-Instruct**, and **Mixtral 8x22B**. We split the resulting dataset into training and test sets in a roughly 9:1 ratio, with a final count of 9,402 training examples and 1,000 test examples.

### A.1.3 DATASET EXAMPLE

Below is a sample row from our dataset.

```
{
  "prompt": "What are 5 words that rhyme with cat?",
  "subset": "hh-rlhf-helpful-online",
  "prompt_id": 8325,
  "response_1": {
    "text": "Here are 5 words that rhyme with "cat":
            1. Bat 2. Hat 3. Mat 4. Sat 5. That",
    "model": "cohere/command-r-plus"
  },
  "response_2": {
    "text": "1. Bat, 2. Hat, 3. Mat, 4. Rat, 5. Sat",
    "model": "openai/gpt-4-turbo"
  },
  "response_3": {
    "text": "Sure, here are five words that rhyme with "cat":
            1. Hat 2. Bat 3. Rat 4. Mat 5. Chat",
    "model": "openai/gpt-4o"
  },
  "response_4": {
    "text": "Here are 5 words that rhyme with "cat": 1. Bat 2. Hat
            3. Mat 4. Rat 5. Sat",
    "model": "anthropic/claude-3-opus"
  },
  "response_5": {
    "text": "Here are 5 words that rhyme with "cat": 1. Bat 2. Hat
            3. Mat 4. Sat 5. Flat",
    "model": "anthropic/claude-3-sonnet"
  },
  "response_6": {
    "text": "Here are 5 words that rhyme with "cat": 1. Hat 2. Mat
            3. Sat 4. Fat 5. Bat",
    "model": "meta-llama/llama-3-70b-instruct:nitro"
  },
  "response_7": {
    "text": "Here are five words that rhyme with "cat": * **bat** *
            **hat** * **mat** * **pat** * **sat**",
    "model": "google/gemini-pro-1.5"
  },
  "response_8": {
    "text": "1. Hat 2. Mat 3. Sat 4. Rat 5. Pat",
    "model": "mistralai/mixtral-8x22b-instruct"
  }
}
```

### A.1.4 Reward Models

Finally, to enable the simulation of many diverse preference models, we select 10 reward models from Reward Bench, built on top of popular base models such as Llama3, Mistral, and Gemma. Their model names on Huggingface are:

- hendrydong/Mistral-RM-for-RAFT-GSHF-v0
- OpenAssistant/oasst-rm-2-pythia-6.9b-epoch-1
- OpenAssistant/oasst-rm-2.1-pythia-1.4b-epoch-2.5
- OpenAssistant/reward-model-deberta-v3-large-v2
- PKU-Alignment/beaver-7b-v1.0-cost
- Ray2333/reward-model-Mistral-7B-instruct-Unified-Feedback
- sfairXC/FsfairX-LLaMA3-RM-v0.1
- weqweasdas/RM-Gemma-2B
- weqweasdas/RM-Gemma-7B
- weqweasdas/RM-Mistral-7B

We evaluate each (prompt, response) pair in the train and test set with each model so that for any personality created by their ensembling, each (prompt, response) pair in the dataset can be scored via a simple weighting.

### A.1.5 Data Records

Each record in our resulting dataset is of the form

$$(x, s, y_1, r_1^{(1)}, ..., r_1^{(k)}, ..., y_l, r_l^{(1)}, ..., r_l^{(k)})$$

where $x$ is an input prompt, $s$ gives the source dataset for the prompt, $y_i$ is a response generated by the LLM indexed by $i$ among our set of $k = 8$, and $r_i^{(j)}$ is the reward score for prompt/response pair $(x, y_i)$ by the reward model indexed by $j$ among a set of $l = 10$ reward models in total.

## A.2 Collection Process

Our prompts were collected by downloading and filtering the open-source datasets mentioned previously. Responses were generated using OpenRouter with a temperature of 1.0 and a maximum token length of 512.

### A.2.1 Model Output License

As of November 14, 2024, our use of model generations as part of a public dataset complies with the terms of use from all the below licenses:

- OpenAI Terms of Use @ https://openai.com/policies/terms-of-use/
- Anthropic Terms of Use @ https://www.anthropic.com/legal/consumer-terms
- Google Terms of Use @ https://ai.google.dev/gemini-api/terms
- Mistral Terms of Use @ https://mistral.ai/terms
- Cohere Term of Use @ https://cohere.com/terms-of-use
- Meta Term of Use @ https://www.llama.com/llama3/license/

## A.3 Preprocessing/cleaning/labeling

All prompts are taken in their exact form from existing open-source datasets, filtered by length according to Appendix A.1.1. LLM responses are not filtered, edited, or cleaned in any way, either for storage or reward scoring.

As a limitation, we note that all prompts and responses have not been manually inspected for quality or safety by a human, although prompts are sourced from existing, reputable datasets, and responses are generated from state-of-the-art language models that have (presumably in the case of black box models) undergone safety alignment. Further, the personas that can be created via ensembling our reward models have not been exhaustively tested for bias or alignment with a particular subgroup of the population.

We also note that there is a known issue with many reward models, such that they may produce different scores under different conditions, in particular when the batch size changes. Our reward scores are produced with a batch size of 1, and are tested for reproducibility and determinism.

### A.4 USES

Our goal in creating the open-source PersonalLLM dataset is to facilitate work on methods to personalize the output of an LLM to the individual tastes of the many diverse users of an application. In our initial paper, we have provided experiments where meta-learning and in-context learning (ICL) are used to leverage an existing user base with interaction histories to improve outcomes for new users. We imagine further work in this direction, as well as potential work on more efficient ways to harness the power of fine-tuning for personalization. Also, in domains like medicine, where privacy is paramount, it may not be possible to include queries from other users in context. Thus, work on privacy-ensuring meta-learning personalization algorithms is needed.

It must be acknowledged that the goal of LLM personalization brings particular risks, including filter bubbles, stereotyping, feedback loops, personification, and manipulation. Given these and many other predictable (and unpredictable) potential risks, it is important that any efforts at LLM personalization are accompanied by research in robust transparency mechanisms and safeguards for personalization algorithms.

### A.5 DISTRIBUTION

### A.5.1 HOSTING

PersonalLLM is available for download on huggingface at `https://huggingface.co/datasets/namkoong-lab/PersonalLLM`.

### A.5.2 LICENSE

We release this dataset under a CC BY-NC 4.0 License, which prohibits commercial use and requires attribution.

### A.6 MAINTENANCE

The authors plan to maintain the dataset. If any instances of dangerous, private, or otherwise undesirable material are found, ICLR is not responsible for any legal violations with respect to the collected benchmark. For correspondence, including requests for data removal, please get in touch at `andrew.siah@columbia.edu` and `tpz2105@columbia.edu`.

## B SIMULATED USER ANALYSIS

### B.1 ADDITIONAL DETAILS

All semantic features are scored using pre-trained models from Huggingface.

- Formality is scored using: s-nlp/roberta-base-formality-ranker
- Educational value is scored using: HuggingFaceFW/fineweb-edu-classifier
- Emotions are scored using: SamLowe/roberta-base-go_emotions

Below is a list of our syntactic features:

- Count of tokens

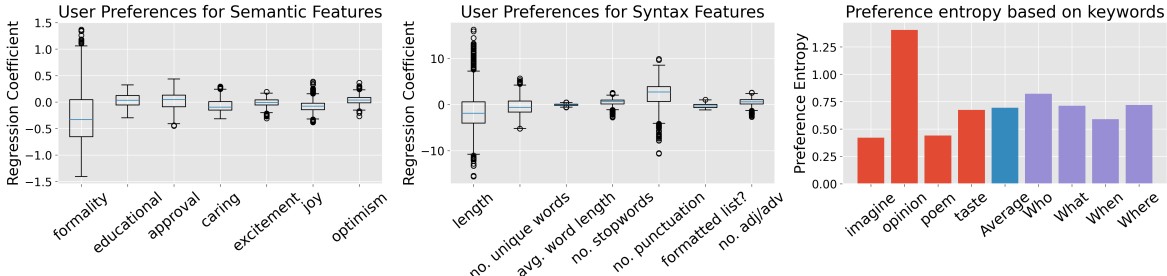

Figure 8: Analysis of persona prompting preferences with respect to prompt and response contents. **Left, middle:** For each user, we train a regression model to predict winning responses based on either semantic (left) or syntactic (middle) features. **Right:** We examine the entropy in population preferences based on keywords in prompts, comparing words we would expect to inspire heterogeneity (e.g., imagine, opinion, poem) to prompts beginning with "who", "when", and "where".

- Count of unique words
- Average word length
- Count of stopwords
- Count of punctuation
- Count of list items (bullets or numbered)
- Count of adjectives and Adverbs

The python package nltk (Bird and Loper, 2004) is used to tokenize, extract stopwords, and tag parts of speech, where necessary.

Our linear regression models are built using sklearn (Pedregosa et al., 2011), with default parameter settings.

Table 3 shows some example questions and answers from the OpinionQA dataset.

Table 3: Example questions and answers from the OpinionQA dataset.

| Question | Answer |
|---|---|
| How worried are you, if at all, about the possibility of using computer programs to make hiring decisions for society as a whole? | [Very worried, Somewhat worried, Not too worried, Not at all worried] |
| Do you think men and women are basically similar or basically different when it comes to their hobbies and personal interests? | [Men and women are basically similar, Men and women are basically different] |

### B.2 ADDITIONAL RESULTS

Tables 4 and 5 show representativeness scores for our PersonalLLM users as well as a selection of LLMs across all 60 demographic groups in the original OpinionQA (Santurkar et al., 2023) study.

Figure 8 examines the preferences of the persona-prompted baseline with respect to semantics, syntax, and keywords, for comparison to Figure 5 which features the same experiments performed on our simulated users. Unlike our simulated users, these simple features are more strongly predictive of response choices, and the presence of subjective cues in prompts lead to a more diverse set of preferences than those seeking simple entity or date responses

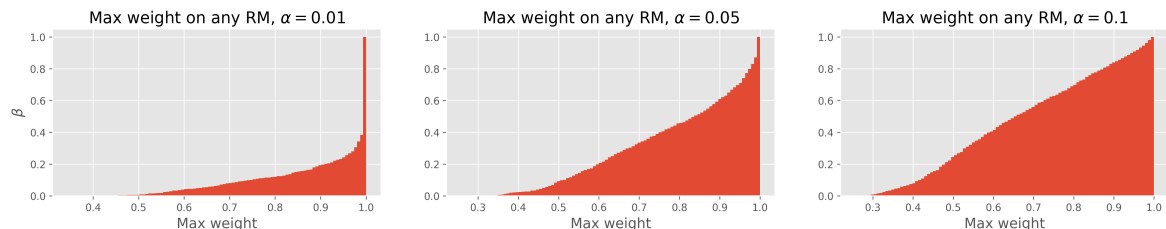

Figure 9: Effects of sampling parameter alpha on the concentration of sampled preference models around a single reward model. As alpha decreases, more weight is put on a single model. As alpha increases, weights are more evenly distributed across reward models.

## C EXPERIMENTS

### C.1 PSEUDOCODE

Below is the pseudocode for the baselines in Section 4. Actual code is available at

---

**Algorithm 1** MetaLearnKShotICLAlgorithm

---

1: **procedure** GENERATERESPONSE(text_user, text_prompt)
2:     similar_users ← FindSimilarUsers(text_user)
3:     similar_prompts ← FindSimilarPrompts(text_prompt, similar_users)
4:     icl_examples ← {}
5:     **for** prompt **in** similar_prompts **do**
6:         winning, losing ← FindWinningLosingResponses(prompt)
7:         icl_examples.append(prompt, winning, losing)
8:         **if** $|$icl_examples$| = k$ **then**          ▷ k is the number of shots
9:             **break**
10:        **end if**
11:     **end for**
12:     final_prompt ← ConstructPrompt(icl_examples, test_prompt)
13:     response ← GenerateLLMResponse(final_prompt)
14:     **return** response
15: **end procedure**

---

## C.2 PROMPT TEMPLATE

Below is a prompt template we used in our experiments for winning and losing responses appended during inference.

---

```
Below are some examples of past conversations with liked and disliked responses
per prompt.

User: {ICL_Prompt_1}
Liked Response: {Prompt_1_Liked_Response}
Disliked Response: {Prompt_1_Disliked_Response}

User: {ICL_Prompt_2}
Liked Response: {Prompt_2_Liked_Response}
Disliked Response: {Prompt_2_Disliked_Response}

Use the contexts above to generate a good response for the user prompt below.
Your response should be similar to the winning responses and dissimilar from
the losing responses.

User: {Test_prompt}
Response:
```

---

Below is a prompt template we used in our experiments for winning responses only appended during inference.

---

```
Below are some examples of past conversations with liked responses per prompt.

User: {ICL_Prompt_1}
Liked Response: {Prompt_1_Liked_Response}

User: {ICL_Prompt_2}
Liked Response: {Prompt_2_Liked_Response}

Use the contexts above to generate a good response for the user prompt below.
Your response should be similar to the winning responses.

User: {Test_prompt}
Response:
```

---

Table 4: Representativeness scores in relation to real human opinions from important demographic groups for different LLMs, as well as our PersonalLLM population.

| Demographic | AI21 Labs | | | OpenAI | PersonalLLM |
| | j1-jumbo | j1-grande-v2 | ada | text-davinci-003 | **Ours** |
| --- | --- | --- | --- | --- | --- |
| Northeast | 0.811 | 0.802 | 0.819 | 0.704 | 0.838 |
| Midwest | 0.810 | 0.797 | 0.820 | 0.701 | 0.833 |
| South | 0.818 | 0.805 | 0.827 | 0.696 | 0.835 |
| West | 0.813 | 0.802 | 0.821 | 0.704 | 0.839 |
| 18-29 | 0.818 | 0.808 | 0.828 | 0.700 | 0.840 |
| 30-49 | 0.814 | 0.804 | 0.823 | 0.702 | 0.837 |
| 50-64 | 0.809 | 0.797 | 0.818 | 0.696 | 0.830 |
| 65+ | 0.792 | 0.779 | 0.800 | 0.699 | 0.818 |
| Male | 0.814 | 0.802 | 0.826 | 0.697 | 0.837 |
| Female | 0.810 | 0.800 | 0.816 | 0.702 | 0.833 |
| Less than high school | 0.828 | 0.812 | 0.835 | 0.685 | 0.832 |
| High school graduate | 0.816 | 0.799 | 0.826 | 0.691 | 0.832 |
| Some college, no degree | 0.814 | 0.804 | 0.823 | 0.701 | 0.836 |
| Associate's degree | 0.811 | 0.800 | 0.821 | 0.700 | 0.834 |
| College graduate | 0.802 | 0.794 | 0.810 | 0.710 | 0.833 |
| Postgraduate | 0.794 | 0.789 | 0.800 | 0.717 | 0.831 |
| Citizen - Yes | 0.814 | 0.802 | 0.823 | 0.700 | 0.836 |
| Citizen - No | 0.816 | 0.812 | 0.818 | 0.706 | 0.833 |
| Married | 0.810 | 0.799 | 0.819 | 0.699 | 0.832 |
| Divorced | 0.809 | 0.796 | 0.817 | 0.696 | 0.830 |
| Separated | 0.814 | 0.801 | 0.818 | 0.694 | 0.830 |
| Widowed | 0.800 | 0.785 | 0.807 | 0.694 | 0.819 |
| Never been married | 0.819 | 0.808 | 0.828 | 0.700 | 0.841 |
| Protestant | 0.810 | 0.797 | 0.820 | 0.694 | 0.828 |
| Roman Catholic | 0.816 | 0.806 | 0.823 | 0.702 | 0.835 |
| Mormon | 0.789 | 0.777 | 0.802 | 0.696 | 0.819 |
| Orthodox | 0.773 | 0.762 | 0.781 | 0.693 | 0.803 |
| Jewish | 0.792 | 0.785 | 0.800 | 0.707 | 0.824 |
| Muslim | 0.794 | 0.788 | 0.792 | 0.697 | 0.816 |
| Buddhist | 0.782 | 0.777 | 0.783 | 0.709 | 0.821 |
| Hindu | 0.796 | 0.794 | 0.789 | 0.707 | 0.816 |
| Atheist | 0.774 | 0.771 | 0.784 | 0.714 | 0.822 |
| Agnostic | 0.785 | 0.781 | 0.794 | 0.717 | 0.828 |
| Other | 0.794 | 0.790 | 0.801 | 0.703 | 0.824 |
| Nothing in particular | 0.815 | 0.802 | 0.824 | 0.700 | 0.839 |
| Rel. attend -  1x/week | 0.807 | 0.793 | 0.816 | 0.690 | 0.824 |
| Rel. attend - 1x/week | 0.811 | 0.798 | 0.819 | 0.696 | 0.829 |
| Rel. attend - 1-2x/month | 0.818 | 0.807 | 0.825 | 0.699 | 0.833 |
| Rel. attend - Few/year | 0.817 | 0.809 | 0.824 | 0.705 | 0.837 |
| Rel. attend - Seldom | 0.811 | 0.800 | 0.821 | 0.703 | 0.835 |
| Rel. attend - Never | 0.806 | 0.795 | 0.816 | 0.701 | 0.836 |
| Republican | 0.791 | 0.776 | 0.805 | 0.680 | 0.812 |
| Democrat | 0.800 | 0.795 | 0.804 | 0.719 | 0.834 |
| Independent | 0.812 | 0.801 | 0.821 | 0.701 | 0.838 |
| Other | 0.820 | 0.804 | 0.832 | 0.693 | 0.839 |
| Less than $30,000 | 0.828 | 0.813 | 0.833 | 0.693 | 0.838 |
| $30,000-$50,000 | 0.814 | 0.802 | 0.822 | 0.698 | 0.834 |
| $50,000-$75,000 | 0.807 | 0.796 | 0.816 | 0.703 | 0.833 |
| $75,000-$100,000 | 0.800 | 0.791 | 0.811 | 0.705 | 0.829 |
| $100,000 or more | 0.797 | 0.790 | 0.807 | 0.708 | 0.831 |

Table 5: Representativeness scores in relation to real human opinions from important demographic groups for different LLMs, as well as our PersonalLLM population.

| | AI21 Labs | | | OpenAI | PersonalLLM |
|---|---|---|---|---|---|
| Demographic | j1-jumbo | j1-grande-v2 | ada | text-davinci-003 | **Ours** |
| Very conservative | 0.797 | 0.778 | 0.811 | 0.662 | 0.811 |
| Conservative | 0.796 | 0.780 | 0.810 | 0.684 | 0.817 |
| Moderate | 0.814 | 0.804 | 0.822 | 0.706 | 0.838 |
| Liberal | 0.792 | 0.788 | 0.799 | 0.721 | 0.833 |
| Very liberal | 0.785 | 0.782 | 0.791 | 0.712 | 0.825 |
| White | 0.807 | 0.794 | 0.817 | 0.699 | 0.832 |
| Black | 0.820 | 0.812 | 0.823 | 0.702 | 0.833 |
| Asian | 0.814 | 0.806 | 0.819 | 0.708 | 0.839 |
| Hispanic | 0.820 | 0.810 | 0.824 | 0.706 | 0.839 |
| Other | 0.801 | 0.783 | 0.807 | 0.681 | 0.818 |

