# OpenReview forum: "PersonalLLM: Tailoring LLMs to Individual Preferences"
_ICLR.cc/2025/Conference — ICLR 2025 Poster_

### Official Review · Reviewer_oPDc · 2024-11-01

**Soundness:** 3
**Presentation:** 2
**Contribution:** 2
**Rating:** 6
**Confidence:** 4

**Summary:**

This paper presents a new dataset of simulated preferences. The data consists of 10K prompts X 8 responses from different LLMs for each prompt X 10 rewards from different reward models. 1000 simulated users are sampled, where each user’s preferences are defined by a weighted sum of rewards (the weights are sampled from a Dirichlet distribution). The data is then used in in-context learning (ICL) for improving the LLM responses w.r.t. the user’s preferences.

Personalization is achieved by ICL, adding examples of good/bad responses according to the weighted reward. The results (Figure 6 left) show that using ICL with historical preferences can improve performance compared to zero-shot.

Learning across users is proposed, retrieving other users with similar preferences from a set of simulated users, and using their preferences for ICL. The results (Figure 6 right) show a small improvement when using both positive and negative preferences compared to ICL using only the user’s history.

**Strengths:**

* The large-scale dataset could be useful for LLM development.
* The alternative to persona-based simulated users seems novel.

**Weaknesses:**

* It is stated in the paper that the goal is not to match preferences of a distribution of real users, but rather to generate diverse preferences that are more heterogeneous/diverse. I think that this requires more justification since random preferences would give even higher diversity but may not be useful.
* Clarity/readability could be improved (see detailed questions).

**Questions:**

* In line 76 it says “in doing so we are able to simulate an entire user base”. On the other hand it says in line 102 that “We do not claim our simulated personal preference models provide a high-fidelity depiction of human behavior”, so this may be a bit confusing and you may want to rephrase these statements. After reading the first one I was hoping for some evaluation of how realistic the simulated users are. This is actually done in “Comparison to Human Preferences” in Section 3, so I guess you are doing some of that? If the goal is to obtain high *coverage* rather than matching the distribution of users, perhaps this can be made explicit and possibly evaluated against real user behavior? Perhaps some measure of support instead of Wasserstein? It would be also interesting to compare the results in Figure 5 to those from standard personas baselines.
Actually, if the goal is coverage then random preferences should give better coverage, but are probably not very useful, so just optimizing coverage doesn’t seem to be a good objective.
Can you please clarify the objective here?
* Another potentially interesting baseline is to have each user choose one of the rewards, a hard choice instead of a weighted sum. There will only be 10 user “types”, so it may be interesting to see how the results change in that case.
* Sometimes there are long multi-line sentences that could be simplified to improve readability and flow. It is easier to read a paper that has no sentences that span more than 2 lines. Some examples:
  * “Given the expected data sparsity in this setting, beyond a particular user’s data, such personalized language systems will likely also rely on historical data from other (similar) users to learn how to learn from a small set of new user feedback (see Figure 2).” Could be simplified/broken (by an LLM): “These personalized language systems will likely use more than just one user's data due to the expected data sparsity in this setting. They will also depend on historical data from other similar users. This helps them learn effectively from a small amount of new user feedback (see Figure 2 for more details).”
  * “We do not claim our simulated personal preference models provide a high-fidelity depiction of human behavior, but rather offer a challenging simulation environment that provides the empirical foundation for methodological innovation in capturing the complex array of human preferences that arise in practice.” Could be made easier to read (by an LLM): “We don't claim that our simulated personal preference models perfectly mimic human behavior. Instead, they offer a challenging simulation that provides a basis for developing new methods. This helps in better capturing the complex range of human preferences encountered in real life.”
  * “While human evaluation like that of Kirk et al. (2024) is a gold standard, wherein fine-grained preference feedback is gathered from a representative sample of diverse and multicultural participants, it is impractical or even impossible to get this feedback throughout the methodology development cycle, meaning that synthetic personal preference models will ultimately be needed.” I had to read this one slowly a couple of times…
  * Line 354: “Two first-order problems…” can be losslessly simplified to “Two problems…”.
* Line 254: choosing only 500 personas may be too little if the goal is to achieve heterogeneity, especially since 1000 users are sampled for PersonalLLM. Can you please include results with 1000 personas? It may actually be interesting to see how the results change when increasing the sample size for both persona and PersonalLLM.
* Line 257: “we can see that the top response receives a majority user vote for only about half of the prompts, while that figure is closer to 90% for the persona prompting baseline.” Sorry, I could not read that from the figure, can you please explain how the results show this?
Also in line 258: “Also, for roughly 60% of prompts, at least 5 different answers are chosen as the best by at least 1 under our set of personas; for LLM persona prompting, it is roughly 30%.” Please explain.
* Line 274: “With respect to changes across the left 3 columns, we can observe that as α increases, preferences become more uniform. However, if α is set too low, user preferences cluster very tightly around the base reward models; we observe this behavior for α = 0.01.” — looking at the figure, it actually seems like there is not much difference between the first 3 columns. Is there a better way to show this difference?
* Line 294: “In Figure 5 (right), we compare the entropy in the population preferences over the responses to a given prompt based on keywords, comparing words we would expect to inspire heterogeneity (e.g., imagine, opinion, poem) to prompts beginning with “who”, “when”, and “where”, which evoke more objective answers.” This was not clear to me, maybe add a formal definition and/or an equation for the entropy? Also, how do standard personas compare to the proposed approach in this task?
* In Section 4.2, is it mentioned how response (and prompt) embeddings are computed?

Minor/typos:
* Line 32: Christiano et al., 2017, not 2023
* In Figure 6 (left), the dashed line is missing from the legend. I am guessing this is the zero-shot performance.

---

> ### Author Response · Authors · 2024-11-19
> **Author Response**
>
> We sincerely appreciate the reviewer for the time and care taken in reviewing our submission and offering feedback on how we might improve our paper. We are encouraged that they agree with the value and novelty of our method of simulating diverse personas for methodological development.  Below, we respond to your particular comments. Noted changes can be viewed in our updated submission PDF.
>
> **It is stated in the paper that the goal is not to match preferences of a distribution of real users, but rather to generate diverse preferences that are more heterogeneous/diverse. I think that this requires more justification since random preferences would give even higher diversity but may not be useful.**
>
> We agree with the reviewer that diversity in and of itself is not the only important criteria for a simulated benchmark.  By building our simulated personas on top of reward models trained with human preference feedback, our simulated reward models inherit some reasonable biases about human preferences, while still exhibiting the desired diversity.  We believe that our analysis in Section 3 shows that our simulated users achieve a good tradeoff between offering reasonable representations of human preferences while overcoming the technical bottleneck in creating diverse preference targets.
>
> **In line 76 it says “in doing so we are able to simulate an entire user base”. On the other hand it says in line 102 that “We do not claim our simulated personal preference models provide a high-fidelity depiction of human behavior”, so this may be a bit confusing and you may want to rephrase these statements. After reading the first one I was hoping for some evaluation of how realistic the simulated users are. This is actually done in “Comparison to Human Preferences” in Section 3, so I guess you are doing some of that?**
>
> We appreciate this concern, and have updated Line 76 to more clearly reflect the nature of our preference models.
>
> **If the goal is to obtain high coverage rather than matching the distribution of users, perhaps this can be made explicit and possibly evaluated against real user behavior? Perhaps some measure of support instead of Wasserstein? It would be also interesting to compare the results in Figure 5 to those from standard personas baselines. Actually, if the goal is coverage then random preferences should give better coverage, but are probably not very useful, so just optimizing coverage doesn’t seem to be a good objective. Can you please clarify the objective here?**
>
> We adopted the methodology of (Santurkar et al., 2023) for evaluating our simulated user base on OpinionQA, in order to measure how well human preferences are represented by our simulated users.  We felt that this made for the strongest basis of comparison, and also allowed including the baseline results from other LLMs without reproducing outputs.  With respect to coverage, this was roughly our goal in evaluating across the 60 demographic groups. We aimed to ensure that the preferences exhibited by our simulated users were reasonable with respect to many different segments of the population, and found positive results.  In consideration of the reviewer’s concern, we will attempt to expand these comparisons before the camera-ready version.
>
> Regarding Figure 5, based on the reviewer’s suggestion we have extended these experiments to the persona prompting baseline.  These new results can be seen in Figure 8.
>
> **Another potentially interesting baseline is to have each user choose one of the rewards, a hard choice instead of a weighted sum. There will only be 10 user “types”, so it may be interesting to see how the results change in that case.**
>
> We agree that this is an interesting setting, and may reflect many applications where users exist in tight “clusters”. We have clarified in line 277 that this can be achieved by lowering the alpha parameter for the Dirichlet distribution for the sampling of weightings.
>
> **Sometimes there are long multi-line sentences that could be simplified to improve readability and flow. It is easier to read a paper that has no sentences that span more than 2 lines.**
>
> We have shortened these sentences, and others that we felt were too long. Thank you for this suggestion.
>
> **Line 254: choosing only 500 personas may be too little if the goal is to achieve heterogeneity, especially since 1000 users are sampled for PersonalLLM. Can you please include results with 1000 personas? It may actually be interesting to see how the results change when increasing the sample size for both persona and PersonalLLM.**
>
> Based on this concern, we have updated Figure 4 (and Figure 8) with results from 1,000 randomly sampled personas, to make for a better comparison with the PersonalLLM simulated user population.

---

> > ### Author Response · Authors · 2024-11-19
> > **Author Response (continued)**
> >
> > **Lack of clarity in results stated in lines 257, 258, and 274.**
> >
> > We have rewritten this paragraph to make it easier to observe the findings that we state.  Also, we have added a new figure to clarify the point about how changes in alpha affect a simulated user base (line 257, 988).
> >
> > **Line 294: “In Figure 5 (right), we compare the entropy in the population preferences over the responses to a given prompt based on keywords, comparing words we would expect to inspire heterogeneity (e.g., imagine, opinion, poem) to prompts beginning with “who”, “when”, and “where”, which evoke more objective answers.” This was not clear to me, maybe add a formal definition and/or an equation for the entropy? Also, how do standard personas compare to the proposed approach in this task?**
> >
> > We have updated this to clarify that we use the standard Shannon entropy (line 297) to measure the entropy in the distribution of preferences over the responses.  Also, in response to the reviewer’s request, we have performed the experiments from Figure 5 on the persona prompting baseline. These results are shown in Figure 8 (line 935).
> >
> > **In Section 4.2, is it mentioned how response (and prompt) embeddings are computed?**
> >
> > Our method for extracting text embeddings is noted in Section 4 lines 359-360, and user embeddings are explained in Section 4.2 lines 443-446.  We have also added a word to line 359 to clarify the point with respect to all text embeddings.

---

> ### Author Response · Authors · 2024-11-22
> **Author Follow-up**
>
> Hello, as the discussion period will be ending in a few days, we wanted to follow up and see if there are any remaining questions we can answer or any other changes we can make to address the reviewer’s concerns. Thank you again for the time and consideration.

---

> > ### Author Response · Authors · 2024-12-01
> >
> > We thank the reviewer again for taking the time to offer feedback on our paper.  As the extended discussion period ends tomorrow, we hope that the reviewer might consider our answers to their questions, as well as the changes that we have made to our submission in response to their concerns.  We would also be happy to respond to any remaining concerns or questions.  Thank you!

---

> ### Comment · Reviewer_oPDc · 2024-12-03
>
> Thank you for the clarifications.

---

> > ### Author Response · Authors · 2024-12-03
> >
> > Thank you very much for considering our rebuttals.  If our responses and updated submission have sufficiently addressed your concerns, we ask that you might consider raising your score.

---

### Official Review · Reviewer_ndCs · 2024-11-02

**Soundness:** 3
**Presentation:** 3
**Contribution:** 4
**Rating:** 8
**Confidence:** 3

**Summary:**

This paper builds a dataset of open-ended prompts and high-quality responses where users might be expected to have different preferences, a method of sampling direct different user preferences based on reward models, and proposes different algorithms for personalization using data across multiple users. In addition, they empirically validate that their proposed method of sampling user preferences beats a baseline persona-based method for generating diverse user preferences.

**Strengths:**

Originality: The paper proposes (as far as I know) an original method for generating diverse user preferences.
Quality: The paper both creates a high-quality dataset, as well as empirically validates that its methodology creates diverse preferences at least as diverse as a persona-based method.
Clarity: The paper is clearly written.
Significance: The paper establishes a dataset and methodology for generating diverse user preferences, which is very important for studying LLM personalization.

**Weaknesses:**

1) The paper uses reward models from a leaderboard (as opposed to fine-tuning to persona data or something), which means that the reward models are all high-quality, but may result in reward models which are less distinct from each other than they might otherwise be. The paper clearly justifies this as not preventing their resampling method from reaching higher diversity than persona-based prompting, but are there other sources of high quality reward functions that might be more different from each other?
2) Similarly, were the leading LLMs used to sample the 8 preferences prompted with personas? The different LLMs might be somewhat more similar to each other than they need to be, but of course resampling the dataset could be quite expensive, and the dataset is quite valuable as is.

**Questions:**

1) Are there other sources of high-quality reward functions that can be used?
2) Were the leading LLMs used to sample the 8 preferences prompted with personas?

**Details Of Ethics Concerns:**

It's worth it to double-check that including the LLM responses in a dataset is within the relevant terms of use -- my impression is that generally they are, but it should be double-checked.

---

> ### Author Response · Authors · 2024-11-19
> **Author Response**
>
> We thank the reviewer for the time and consideration offered in reviewing our paper.  We are encouraged to see that they see our work as original and clearly presented, and that they feel our testbed may make a significant contribution to the field of LLM personalization.  Below we respond to particular points of feedback.
>
> **The paper uses reward models from a leaderboard (as opposed to fine-tuning to persona data or something), which means that the reward models are all high-quality, but may result in reward models which are less distinct from each other than they might otherwise be. The paper clearly justifies this as not preventing their resampling method from reaching higher diversity than persona-based prompting, but are there other sources of high quality reward functions that might be more different from each other?**
>
> We agree with the reviewer that it is worth considering other methods for producing a large and diverse set of high-quality reward functions, given the acknowledged shortcomings of our approach.  We are not aware of any such methods at this time but hope that researchers take inspiration from this work and are able to develop more faithful and diverse simulators in the future.  We have further acknowledged this concern and the need for further work in the “Future Directions” section of our paper.
>
> **Similarly, were the leading LLMs used to sample the 8 preferences prompted with personas? The different LLMs might be somewhat more similar to each other than they need to be, but of course resampling the dataset could be quite expensive, and the dataset is quite valuable as is.**
>
> We hit our budget constraint as an academic lab producing the LLM responses for our dataset, and were not able to probe the effects of persona prompting on these responses from these strong models. We agree that this would be a very interesting direction for future research, and hope this is enabled by the release of our dataset and pipeline code.
>
> **It's worth it to double-check that including the LLM responses in a dataset is within the relevant terms of use -- my impression is that generally they are, but it should be double-checked.**
>
> Thank you for this suggestion. We checked this before submission but did not explicitly state this in our dataset card in the appendix. We have added this explicitly in Section A.2.1.

---

> > ### Author Response · Authors · 2024-11-22
> > **Author Follow-up**
> >
> > Hello, as the discussion period will be ending in a few days, we wanted to follow up and see if there are any remaining questions we can answer or any other changes we can make to address the reviewer’s concerns. Thank you again for the time and consideration.

---

> ### Comment · Reviewer_ndCs · 2024-11-26
> **Thanks!**
>
> Thanks for double-checking the terms of use! I stand by my positive assessment, and appreciate the authors explaining their constraints.

---

### Official Review · Reviewer_YmDG · 2024-11-03

**Soundness:** 2
**Presentation:** 2
**Contribution:** 2
**Rating:** 5
**Confidence:** 5

**Summary:**

The paper aims to propose a dataset called PERSONALLLM for the personalization AI area, which contains users’ preference illustrated by a prompt with eight responses. Specifically, the user responses are built up by various LLMs, e.g., GPT4, Claude 3.

The authors then propose in-context learning and meta-learning methods as baselines for two scenarios from PERSONAL. The results show that there is much room for improvement in solving the personalization problem in the proposed PERSONAL.

**Strengths:**

1. The paper uses multiple LLMs to generate various responses to improve the confidence of dataset.
2. The paper provides a specific analysis of the dataset.

**Weaknesses:**

1. The paper is unclear about what the preference is in the data; is it user preference of items in recommender systems or any replacement of NLP tasks or others?
2. The paper is uclear about how the PERSONALLLM is formulated, the author presented the reward model, but how it is trained/built up.
3. The author illustrates the heter preference PERSONALLLM involves in which differs from the home ones, but how these two preferences demonstrate is not clear.

**Questions:**

Answering and solving the weakness questions clearly can greatly help the reviewer target the focus of the paper. For the reviewer, these issues require a lot of time to carefully polish the paper before they can be completed.  In addition, the review would ask:
What is the relationship between PERSONALLLM and recommender system? Is it a replacement of existing ones or a more general preferenc-based system incuding RS? Why?

---

> ### Author Response · Authors · 2024-11-19
> **Author Response**
>
> We thank the reviewer for their consideration of and feedback on our submission.  Please see below for responses to specific questions and comments.
>
> **The paper is unclear about what the preference is in the data; is it user preference of items in recommender systems or any replacement of NLP tasks or others?**
>
> Our testbed is meant to explore user preferences with respect to different possible LLM responses to a user query.  We have attempted to convey this in the first 3 paragraphs of the introduction, especially lines 40-46, as well as Figures 1-3. If the reviewer has any further suggestions to clarify this point we would be happy to update the submission.
>
> **The paper is uclear about how the PERSONALLLM is formulated, the author presented the reward model, but how it is trained/built up.**
>
> We detail our approach to producing simulated personal reward models in Section 2.2.  To summarize, we address the challenge of simulating diverse user preferences using a set of strong, open-source RLHF reward models (sourced through RewardBench https://huggingface.co/spaces/allenai/reward-bench). We generate simulated users by sampling weighted combinations of these models, defining user-specific preferences as weighted sums of reward scores from a selected set of 10 high-performing models, such as Llama3 and Mistral. This approach enables scoring of any (prompt, response) pair through simple weightings, providing a scalable framework for studying personalization.
>
> **The author illustrates the heter preference PERSONALLLM involves in which differs from the home ones, but how these two preferences demonstrate is not clear.**
>
> One of our main goals in creating PersonalLLM was to create a set of preference models and data such that the preference models would offer heterogeneous preferences over the responses in the data.  In order to verify our approach, in Section 3 we examine whether populations of personal preference models sampled via the method outlined in Section 2.2 do in fact display heterogeneous preferences over the prompt/response pairs in our dataset, and compare to the popular persona prompting baseline.  In Figure 4 and the resulting analysis, we find that our method produces heterogeneous preferences over our dataset of prompts and responses, considerably more so than persona prompting an LLM.  For example, under our method the most popular response to a query receives a majority user vote for only about half of the prompts, while that figure is closer to 90% for the persona prompting baseline.  Also, for roughly 60% of prompts, at least 5 different answers are chosen as the best by at least 1 user under our set of simulated preference models; for LLM persona prompting, it is roughly 30%, meaning that for most data examples, at least half of potential responses are not preferred by a single user.  Finally, our ensembled preference models have a much more diffuse set of preferences over the response-generating LLMs than persona prompting.
>
> **What is the relationship between PERSONALLLM and recommender system? Is it a replacement of existing ones or a more general preferenc-based system incuding RS? Why?**
>
> Our work on PersonalLLM is inspired by classic recommender systems in several ways. First, we aim for PersonalLLM to allow for the simulation of a large number of users, enabling the study of the full personalization paradigm for applications such as search engines and recommender systems wherein a historical database of user data is leveraged to personalize new interactions.  We also build on the existing paradigm of using simulated rewards for developing recommender systems.  Further, the setting in Section 4.2 resembles typical
> recommendation systems, but “actions” are now defined over the space of natural language outputs instead of a fixed set of items.  We attempt to highlight this throughout the submission, but we will make sure to emphasize it further in the camera-ready version.

---

> > ### Author Response · Authors · 2024-11-22
> > **Author-follow**
> >
> > Hello, as the discussion period will be ending in a few days, we wanted to follow up and see if there are any remaining questions we can answer, or any other changes we can make to address the reviewer’s concerns. Otherwise, we hope that the reviewer may consider raising their score. Thank you again for the time and consideration.

---

> > > ### Author Response · Authors · 2024-12-01
> > >
> > > We thank the reviewer again for taking the time to offer feedback on our paper.  As the extended discussion period ends tomorrow, we hope that the reviewer might consider our rebuttal and please let us know if there are any remaining questions or concerns that we might be able to address.  Thank you!

---

### Official Review · Reviewer_kUTu · 2024-11-04

**Soundness:** 2
**Presentation:** 2
**Contribution:** 2
**Rating:** 5
**Confidence:** 4

**Summary:**

The paper introduces PersonalLLM, a public benchmark designed to personalize Large Language Models (LLMs) to better align with individual user preferences. The benchmark focuses on simulating diverse personal preferences using a set of pre-trained reward models. The dataset consists of open-ended prompts paired with multiple high-quality LLM responses, and the goal is to optimize personalization by leveraging historical user data. Basic baselines, including in-context learning and meta-learning, are explored to showcase the utility of this benchmark, setting the stage for future research into personalization algorithms for LLMs.

**Strengths:**

1. PersonalLLM provides a way to enhance the personalization of LLMs, which is an impactful direction to enhance the user experience.

2. The benchmark includes extensive open-ended prompts with responses from state-of-the-art LLMs.

3. The paper highlights the use of meta-learning to address data sparsity issues by leveraging historical interactions, which is crucial for real-world applications where personalized models lack sufficient user-specific data.

**Weaknesses:**

1. The personal preference models used to simulate diverse user preferences are not convincing enough to represent real users. First, it is difficult to verify whether the linear combination of scores from reward models aligns with the distribution of user rewards in the real world. Second, the candidate responses generated by LLMs may not cover real-world user-specific responses, making it challenging for LLMs to learn user-specific preferences or align with user-specific backgrounds. For instance, users may have particular preferences or habits that general reward models inherently struggle to account for when providing accurate rewards.

2. The paper lacks an overarching figure that illustrates the construction logic of the dataset and what the samples within the dataset look like.

3. The comparison of the paper with other relevant personalized LLM benchmarks, such as the LaMP dataset.

4. Some related concepts are not clearly explained, such as 'interaction history', 'preference data', and 'user data,' which are not well defined.

**Questions:**

see the weakness

---

> ### Author Response · Authors · 2024-11-19
> **Author Response**
>
> We thank the reviewer for the time and care taken in reviewing our submission.  We are encouraged that they felt we had taken an impactful direction, and that they recognized the importance of using meta-learning to address the data sparsity issue in these settings.  Below, we respond to particular points of feedback.
>
> **The personal preference models used to simulate diverse user preferences are not convincing enough to represent real users. First, it is difficult to verify whether the linear combination of scores from reward models aligns with the distribution of user rewards in the real world. Second, the candidate responses generated by LLMs may not cover real-world user-specific responses, making it challenging for LLMs to learn user-specific preferences or align with user-specific backgrounds. For instance, users may have particular preferences or habits that general reward models inherently struggle to account for when providing accurate rewards.**
>
> Our goal is not to produce a fully realistic simulation of human behavior, but instead to create a challenging simulation environment that can serve as an empirical foundation for innovation in LLM personalization. In this case, this primarily means creating a diverse enough set of preference models such that different users prefer different responses.  In Section 3 we show that creating diverse preferences is challenging with existing approaches and that we resolve this technical bottleneck with our simulation method.  Given that our testbed is the first to enable the study of settings where a large historical database of user data can be leveraged to personalize new chat outputs for new users, we believe that PersonalLLM represents a meaningful contribution towards advancing the personalization of language-based agents.
>
> **The paper lacks an overarching figure that illustrates the construction logic of the dataset and what the samples within the dataset look like.**
>
> We appreciate the reviewer pointing this out, and have added a new Figure 7 that illustrates the construction logic of the dataset.  Further, we have added an example of what a data sample looks like to Appendix A.
>
> **The comparison of the paper with other relevant personalized LLM benchmarks, such as the LaMP dataset.**
>
> Thank you for pointing this out.  We have added LaMP, as well as other relevant LLM personalization benchmarks, to our related works section.
>
> **Some related concepts are not clearly explained, such as 'interaction history', 'preference data', and 'user data,' which are not well defined.**
>
> Thank you for the suggestion, we have attempted to clarify these terms in the paper.

---

> > ### Author Response · Authors · 2024-11-22
> > **Author Follow-up**
> >
> > Hello, as the discussion period will be ending in a few days, we wanted to follow up and see if there are any remaining questions we can answer or any other changes we can make to address the reviewer’s concerns. Otherwise, we hope that the reviewer may consider raising their score. Thank you again for the time and consideration.

---

> > > ### Author Response · Authors · 2024-12-01
> > >
> > > We thank the reviewer again for taking the time to offer feedback on our paper.  As the extended discussion period ends tomorrow, we hope that the reviewer might consider our rebuttal and please let us know if there are any remaining questions we can answer.  Thank you!

---

### Author Response · Authors · 2024-11-19
**Author Response**

We thank the reviewers for their time and careful feedback. We appreciate that all reviewers highlighted the significance of the personalization problem and the potential for PersonalLLM to make a novel contribution to the field of LLM personalization along several dimensions.

- Motivated by the key methodological gap in personalizing LLMs, we provide an empirical testbed that can spur algorithmic innovations. We release a new open-source dataset with over 10K open-ended prompts paired with 8 high-quality responses from top LLMs scored by 10 different SoTA reward models.
- We propose a novel method for sampling diverse "personas" via randomly weighted ensembles of reward models, to avoid the need for opaque and expensive GPT4o evaluations or unreliable (and possibly discriminatory) persona prompting. Unlike standard approaches, our novel method creates a diverse set of preferences.
- At its core, our work is guided by the belief that the value of a benchmark lies in its capacity to drive methodological progress. We do not claim our personas replicate human behavior—this is a lofty goal and outside the scope of this work. Instead, we aim to create a rigorous and reasonable simulation environment that serves as an empirical foundation for innovation in LLM personalization.
- Our benchmark creates new possibilities for algorithmic development, by providing a challenging enough setting that methodological progress therein can imply progress on real applications. As an analogy, while ImageNet is noisy and synthetic--differentiating between 120 dog breeds is not a realistic vision task--it provides a challenging enough setting that methodological progress on ImageNet implies progress on real applications.  We thus believe PersonalLLM represents a meaningful step forward in advancing the personalization of language-based agents.

Also, we note that we attempted to follow best practices by including a dataset card, to inform users about potential concerns and how to responsibly use the data.  We also discuss the risks and ethical implications of our dataset release in Section 6. If there are any remaining concerns that we can allay here, please let us know.

Below, we respond to each reviewer's individual concerns.  We have also submitted an updated manuscript reflecting reviewers’ concerns.

---

### Meta-Review · Area_Chair_8G5P · 2024-12-21

**Metareview:**

This paper introduces PERSONALLLM, a novel dataset for advancing research in the personalization AI domain. The dataset captures user preferences through prompts accompanied by eight responses generated by various large language models (LLMs), including GPT-4 and Claude 3. To benchmark performance on PERSONALLLM, the authors propose in-context learning and meta-learning methods as baseline approaches for two personalization scenarios. Experimental results reveal significant room for improvement in addressing personalization challenges within the dataset, highlighting the potential for further advancements in this field.

Positive points:
+ This paper introduced PersonalLLM, which is an impactful direction to enhance the user experience.
+ The dataset proposed in the paper is well-analyzed.
+ The paper is well written.

Negative point:
- The simulated dataset can be not reliable (e.g., linear combination of the reward models)
- The comparison of the paper with other relevant personalized are insufficient

**Additional Comments On Reviewer Discussion:**

In the rebuttal period, the authors have addressed most concerns raised by the reviewers. The two negative ratings are both 5. I have read the comments from the reviewers, I believe most of the concerns can be easily addressed in the final version. As a result, I recommend acceptance of the paper.

---

### Decision · Program_Chairs · 2025-01-22

Accept (Poster)